# SARS-CoV-2-Specific Immune Cytokine Profiles to mRNA, Viral Vector and Protein-Based Vaccines in Patients with Multiple Sclerosis: Beyond Interferon Gamma

**DOI:** 10.3390/vaccines12060684

**Published:** 2024-06-19

**Authors:** Georges Katoul Al Rahbani, Christina Woopen, Marie Dunsche, Undine Proschmann, Tjalf Ziemssen, Katja Akgün

**Affiliations:** Center of Clinical Neuroscience, Department of Neurology, Carl Gustav Carus University Hospital, Technical University Dresden, 01307 Dresden, Germany; georges.rahbani@ukdd.de (G.K.A.R.); christina.woopen@ukdd.de (C.W.); marie.dunsche@ukdd.de (M.D.); undine.proschmann@ukdd.de (U.P.); tjalf.ziemssen@ukdd.de (T.Z.)

**Keywords:** multiple sclerosis, immunomodulation, cytokine profile, mRNA vaccines, viral vector vaccines, protein-based vaccines, ocrelizumab, sphingosine 1-phostphate receptor modulators, glatiramer acetate, vaccination strategies

## Abstract

Disease-modifying therapies (DMTs) impact the cellular immune response to severe acute respiratory syndrome coronavirus type 2 (SARS-CoV-2) vaccines in patients with multiple sclerosis (pwMS). In this study, we aim to elucidate the characteristics of the involved antigen-specific T cells via the measurement of broad cytokine profiles in pwMS on various DMTs. We examined SARS-CoV-2-specific T cell responses in whole blood cultures characterized by the release of interleukin (IL)-2, IL-4, IL-5, IL-10, IL-13, IL-17A, interferon-gamma (IFN-γ), and tumor necrosis factor-alpha (TNF-α), as well as antibodies (AB) targeting the SARS-CoV-2 spike protein in pwMS following either two or three doses of mRNA or viral vector vaccines (VVV). For mRNA vaccination non-responders, the NVX-CoV2373 protein-based vaccine was administered, and immune responses were evaluated. Our findings indicate that immune responses to SARS-CoV-2 vaccines in pwMS are skewed towards a Th1 phenotype, characterized by IL-2 and IFN-γ. Additionally, a Th2 response characterized by IL-5, and to a lesser extent IL-4, IL-10, and IL-13, is observed. Therefore, the measurement of IL-2 and IL-5 levels could complement traditional IFN-γ assays to more comprehensively characterize the cellular responses to SARS-CoV-2 vaccines. Our results provide a comprehensive cytokine profile for pwMS receiving different DMTs and offer valuable insights for designing vaccination strategies in this patient population.

## 1. Introduction

Multiple sclerosis (MS) is a chronic immune-mediated disease of the central nervous system (CNS) leading to demyelination and in later stages to neuronal degeneration, and it represents a significant worldwide disease burden [1,2]. Most patients with MS (pwMS) are treated with immune-modulating or immunosuppressive disease-modifying therapies (DMT) [3]. These, however, have been shown to interfere with and even suppress many desirable immune responses such as immune responses against vaccines [4,5,6]. Vaccines against the beta coronavirus severe acute respiratory syndrome 2 (SARS-CoV-2) are of interest because of their widespread use [7]. Previous studies in pwMS have mostly focused on the humoral responses against SARS-CoV-2 messenger RNA (mRNA) and viral vector vaccines (VVV), and have shown that some but not all DMTs affect the antibody (AB) response to those vaccines [8,9,10]. Nearly all pwMS treated with glatiramer acetate (GA) produced an AB response comparable to that of patients that are untreated (UT) and healthy controls after two vaccine doses [9]. However, only around 40% of pwMS on anti-CD20 (aCD20) therapies such as ocrelizumab (OCR) and 75% of those on sphingosine 1-phostphate receptor modulators (S1PR) exhibited seroconversion after receiving SARS-CoV-2 vaccines [8,9,10,11,12]. Studies on T cell immunity generally focused on SARS-CoV-2 antigen (Ag)-specific interferon gamma (IFN-γ) release. GA patients developed a positive IFN-γ release after vaccination, similarly to UT patients, as compared to only 14% of S1PR-treated pwMS. On the other hand, the strongest T cell activation was found in pwMS on aCD20 after vaccination [12,13]. Given the inhomogeneity of SARS-CoV-2-specific IFN-γ T cell responses across DMTs, we posed the question of whether other cytokines could be measured to define the T cellular response to SARS-CoV-2 vaccines.

In this study, we aim to use a multiplex cytokine assay evaluation to measure a broader cytokine profile of the SARS-CoV-2-specific T cells involved in pwMS under treatment with different DMTs after two and three doses of SARS-CoV-2 mRNA or VVV. We also aim to clarify whether other cytokines can better characterize the T cell response to SARS-CoV-2 vaccines in pwMS on DMTs that lead to low or absent IFN-γ release.

## 2. Materials and Methods

Patient selection: A cohort study was carried out among pwMS treated at the MS Center Dresden, Germany. During their routine clinical visits from June 2021 to October 2022, the patients were screened and included in this study according to the following inclusion criteria: confirmed MS diagnosis, age >18 years, UT or receiving treatment with GA, OCR, S1PR (fingolimod (FTY), siponimod (SIP), ponesimod (PON), ozanimod (OZA)), and completed vaccination (two or three doses) with mRNA (BNT162b2, mRNA-1273) or VVV (AZD1222, Ad26.COV2.S). A previous SARS-CoV2 infection was defined by clinical symptoms and positive SARS-CoV2 PCR or COVID-19 Antigen Rapid Test Kit. Three different approaches were chosen for further analyses.

In the first cohort of the study (fully vaccinated cohort, *n* = 126), we aimed to elucidate the cytokine profile and humoral responses in pwMS after two doses of mRNA or VVV. It included UT pwMS, as well as those treated with GA, S1PR, and OCR.

The second cohort (booster cohort, *n* = 28) aimed to assess the B and T cell responses after administration of booster vaccines in previously fully vaccinated patients. This cohort only included pwMS on OCR or S1PR modulators. OCR and S1PR were the DMTs of interest in the booster cohort based on the preliminary results of the fully vaccinated cohort, as well as the available literature [10,13,14,15,16,17,18].

In a third approach, the data of 975 pwMS from the same center were screened for insufficient response to at least two doses of mRNA or VVV. Insufficient immune response was defined as a negative T cell response (IFN-γ release to SARS-CoV-2 Ag1 and Ag 2 < 0.15 IU/mL) and anti-SARS-CoV-2 spike protein IgG AB < 200 BAU/mL. To screen for a sufficient T and B cell response, an ELISA-based SARS-CoV-2 QuantiFERON IFN-γ release assay (Qiagen, Hilden, Germany) and LIAISON^®^ SARS-CoV-2 S1/S2 IgG Chemiluminescent Immunoassay (DiaSorin, Saluggia, Italy) were used, respectively. A total of 167 patients fulfilled the criteria, of which 64 consented to be part of the third cohort (protein-based cohort). These patients were then followed in a prospective longitudinal cohort during which they received two vaccinations with NVX-CoV2373 (at baseline and three weeks later). Blood samples were collected at baseline before vaccination (T0), and follow-up samples were collected at three weeks after the first dose (T1) and at 4–8 weeks post second dose (T2). Table 1 and Table 2 present detailed patient characteristics.

T cell response and cytokine profile measurement: Lithium-heparin blood samples were collected during the patients’ routine clinic visits. The samples were incubated for 16–24 h at 37 °C with either SARS-Cov-2 Ag1 (containing CD4+ epitopes from the S1 subunit of the spike protein), Ag2 (containing CD4+ and CD8+ epitopes from the S1 and S2 subunit of the spike protein), mitogen (M) as a positive control, or a negative control (N) using QuantiFERON SARS-CoV-2 Blood Collection Tubes (Qiagen, Hilden, Germany). Then, the QuantiFERON blood collection tubes were centrifuged, and plasma supernatant was obtained and stored at −80 °C until evaluation. IFN-γ levels were initially measured using ELISA-based SARS-CoV-2 QuantiFERON IFN-γ release assay as mentioned above.

For patients in the fully vaccinated cohort, booster cohort, and those included in the protein-based cohort, we quantified the concentrations of interleukin (IL) 2, IL-4, IL-5, IL-10, IL-13, IL-17A, IFN-γ, and tumor necrosis factor alpha (TNF-α) response to SARS-CoV-2 vaccines. The cytokine concentrations in the obtained plasma were then measured using a cartridge-based Multi-Analyte Automated ELLA assay (ELLA™, Bio-Techne, Minneapolis, MN, USA). The coefficients of variation (CV) for the duplicate measurements were calculated, and measurements of samples with CV > 20% were repeated and then eventually excluded if the desired CV was not reached. Afterwards, the mean of the duplicates was calculated for N, S1-, and S2-stimulated samples, and control values (N) were subtracted from Ag-stimulated values (S1/S2). Since the literature reports a baseline circulating plasma IFN-γ concentration of around <20 pg/mL in healthy individuals, a positive T cell response was defined as an IFN-γ release >20 pg/mL [19,20]. Research concerning concentration thresholds for the other cytokines above which the T cell response would be considered positive was very scarce. Therefore, in our work, we defined positive T cell responses only based on IFN-γ > 20 pg/mL, and independently from the release of other cytokines. The upper limit of quantification (ULOQ) and lower limit of quantification (LLOQ) for each cytokine are ranges whose respective means are the following: IL-2: 2050 and 0.54 pg/mL, IL-4: 1920 and 0.5 pg/mL, IL-5: 1248 and 0.13 pg/mL, IL-10: 2212 and 0.58 pg/mL, IL-13: 13,720 and 3.6 pg/mL, IL-17A: 10,000 and 1.05 pg/mL, IFN-γ: 4000 and 0.17 pg/mL, TNF-α 1160 and 0.3 pg/mL. For statistical analysis, measurements that fell within the range of the LLOQ were set to the mean of that range. No values exceeded the ULOQ.

Measurement of SARS-CoV-2-specific Abs: Anti-SARS-CoV-2 spike protein receptor-binding domain (RBD) IgG ABs were measured using an electrochemiluminescence immunoassay (ECLIA) with a Cobas e801 Immunoassay system (Roche, Basel, Switzerland). The seropositivity cut-off was defined as 0.8 U/mL, as recommended by the manufacturer’s instructions. The lower detection limit was 0.43 U/mL, and values below 0.43 U/mL were set to half the detection limit, at 0.215 U/mL. The upper detection limit was 25.000 U/mL, and values above were set to 25.001 U/mL. The assigned unit U/mL corresponds to the WHO international standard binding AB units (BAU)/mL.

Statistical analysis: All analyses, quantifications, and graphical representations were conducted using IBM SPSS Statistics version 28.0.0.0, GraphPad Prism 9, and Canva.com. For the descriptive analysis, means and standard deviations for the total study sample and each individual cohort were calculated. Shapiro–Wilk tests and Q-Q plots were then created for the classification of the distribution of the outcomes. Correlations were calculated using Spearman’s rank correlation and were reported in terms of Spearman’s Rho coefficient (ρ) and statistical significance.

In the fully vaccinated and booster cohort, for outcomes whose distribution was right-skewed, the release of each individual cytokine as well as AB titers was analyzed using generalized linear models with Tweedie log function and a robust fit. For outcomes whose distribution was binomial, a negative binomial function with a log link was used. For the outcomes that are common between these two cohorts, generalized linear mixed models with Tweedie log function and a robust fit were used in order to assess these outcomes longitudinally. For the protein-based vaccine cohort, generalized linear mixed models with Tweedie Function were also used as this cohort forms a longitudinal cohort. For all models, the outcomes were reported as model estimates (mean and 95% Confidence Interval (CI)). The fixed factors were age, gender, treatment modality, previous infections with SARS-CoV-2, and time interval between vaccination and sampling. Sidak correction for pairwise testing was applied.

## 3. Results

### 3.1. SARS-CoV-2-Specific Immune Responses in Fully Vaccinated Patients

Among the fully vaccinated participants (*n* = 128, Table 1), 10 (7.9%) of our participants were previously infected with SARS-CoV-2 prior to sampling. These cases were not distributed similarly among our different treatment cohorts. Therefore, the data of these patients were excluded for statistical models comparing the different treatment cohorts.

A positive serum anti-SARS-CoV-2 RBD AB titer was observed in 100% (28/28) of UT pwMS, 100% (27/27) of pwMS treated with GA, 81% (26/32) of S1PR patients, and 33% (13/39) of OCR patients. Mean anti-SARS-COV-2 AB titer was highest in GA patients as compared to UT and all other treatment groups (Figure 1a). AB concentration in UT patients was also significantly higher than in OCR and S1PR patients. No statistically significant difference was observed between the OCR and S1PR cohorts.

Furthermore, 56% (15/27) of GA patients, 64% (18/28) of UT patients, 12.5% (4/32) of S1PR, 82% (33/39) of OCR patients presented a positive T cell response, as defined previously by IFN-γ release >20 pg/mL. IL-2 and IFN-γ presented the most prominent SARS-CoV-2-specific release in all investigated groups. No difference in the evaluated cytokine-profiles was observed between UT and GA patients. IL-2, IL-4, IL-5, IL-13, and IFN-γ release was significantly higher in OCR patients but significantly lower in S1PR patients compared to the other treatment groups. IL-10 and IL-17A release did not differ between the UT, GA, S1PR, and OCR cohorts (Figure 1b,c). TNF-α presented consistently high concentrations in all groups. A shorter time interval between vaccination and blood sampling was significantly correlated with increased IL-4, IL-5, and IL-13 release. Increasing age was significantly correlated with decreasing IL-10, IL-13, and IL-17 release. Gender did not have an effect on cytokine concentrations.

With IFN-γ being a well-established marker of cellular immunity, our aim was to analyze the correlation between its release and the release of other cytokines after vaccination. IL-2 release correlated most strongly to IFN-γ release and did so consistently among all treatment groups, with the largest correlation being in UT pwMS (ρ = 0.886, *p* < 0.01) and the smallest correlation in S1PR (ρ = 0.685 *p* < 0.01) (Appendix A). In addition, IL-4, 5, 10, and 13 releases also correlated with IFN-γ release. Among the fully vaccinated participants, regardless of DMT modality, IL-5 correlated strongly (ρ = 0.783, *p* < 0.01), whereas IL-4 (ρ = 0.543, *p* < 0.01), IL-10 (ρ = 0.400, *p* < 0.01), and IL-13 (ρ = 0.464, *p* < 0.01) correlated moderately (Appendix A). This trend was conserved among the different DMT groups except S1PR. IL-17 release did not show any statistically significant correlation with IFN-γ.

### 3.2. SARS-CoV-2-Specific Immune Responses after Booster Vaccination

In the second part of our approach, 28 pwMS were evaluated in the booster cohort (Table 1). All (12/12) of S1PR and 18.75% (3/16) of OCR patients had positive serum anti-SARS-CoV-2 RBD AB titer after vaccination with three doses. AB titers in S1PR patients were found to be significantly higher as compared to the titers of OCR patients (Figure 2a). No statistically significant difference was found when comparing the fully vaccinated with the booster cohorts within each treatment group. Older patients had lower AB titers in both treatment groups (*p* = 0.017), and previous infection with SARS-CoV-2 was significantly associated with an increased AB response (*p* = 0.025). Gender did not have a statistically significant effect on AB titers (*p* = 0.303).

T cell response varied between both cohorts. In total, 81.2% (13/16) of OCR patients but none (0/14) of the S1PR patients had a positive T cell response (IFN-γ release > 20 pg/mL) to Ag1 and Ag2. IL-2, IL-4, IL-5, IL-10, IL13, and IFN-γ were significantly higher in OCR patients compared to S1PR patients with the highest concentrations of IL-2 and IFN-γ (Figure 2b,c). IL-17 did not show any relevant difference between both groups and was consistently low. TNF-α was again characterized by an increase in both groups. We found no statistically significant difference between T cell responses in the booster and in the fully vaccinated cohort. Older patients had lower levels of IL-2, IL-5, IL-10, and IL-13. A shorter interval between vaccination and sampling significantly increased IL-4, IL-13, and IL-17A release. Females had a higher cytokine release of IL-5 and IL-17A, whereas males had a higher cytokine release of IL-4 and TNF-α.

### 3.3. The Protein-Based Vaccine Cohort: Population and Patient Characteristics

In order to assess the benefit of NVX-CoV2373 vaccination in patients who had not responded to mRNA and/or VVV (negative T cell response and anti-SARS-CoV-2 spike protein IgG AB < 200 BAU/mL), 63 patients were recruited to form a prospective longitudinal cohort. Most patients were previously triple-vaccinated, while some had a combination of vaccinations and SARS-CoV-2 infections (Table 2). Three patients reported a SARS-CoV-2 infection between the two doses (T1 and T2) of NVX-CoV2373 administered during this study. These patients were tested before the second dose was administered to assess whether an immune response had developed upon infection. Since all three patients did not mount a sufficient immune response as defined by our inclusion criteria, the second dose of NVX-CoV2373 was applied and the results of these patients were included in the analysis. One patient withdrew from further vaccinations due to reported side effects, and the AB data of seven patients were disregarded because they had received SARS-CoV-2 pre-exposure prophylaxis treatment during the study period.

### 3.4. SARS-CoV-2-Specific Immune Responses after Protein-Based Vaccines

Before the first NVX-CoV-2373 administration at T0, 2.2% (1/45) of S1PR patients, and 11% (2/18) of OCR patients had a positive T cell response (IFN-γ > 20 pg/mL), while no patients had an AB titer greater than the pre-defined cutoff (Figure 3 and Figure 4).

At T1 (3 weeks after the first vaccine dose), the T cell response of S1PR patients did not change. In contrast, 72.2% (13/18) of the OCR patients who had previously not responded to mRNA or VVV showed a positive T cell response (Figure 3a,b). No statistically significant increase in any cytokine was observed in S1PR pwMS (Figure 3c,d), while statistically significant increases in IL-2, IL-4, IL-5, and IFN-γ as compared to T0 were detected in OCR pwMS (Figure 3a,b). At T1, 32% (15/47) of the S1PR patients had a sufficient B cell response, with the AB titer being significantly higher at T1 as compared to T0. As for OCR patients, only 11% (2/18) of patients presented a sufficient AB response, and no significant increase in AB concentrations was observed compared to T0 (Figure 4a).

At T2 (4–8 weeks post second vaccination), no significant T cell recall response was triggered after the second NVX-CoV2373 dose for S1PR patients. Only 6.67% (3/45) had a positive T cell response (IFN-γ > 20 pg/mL) in contrast to 72% (13/18) of OCR patients. Concerning individual cytokines in S1PR pwMS, none were released in significantly higher concentrations compared to T0 or T1 (Figure 3c,d), while OCR patients presented statistically significant increases in IL-2, IL-4, IL-5, and IFN-γ as compared to T0. The humoral response improved for both S1PR and OCR patients as compared to T0, with 62% (28/45) of S1PR patients showing sufficient serum AB titers. In addition, 22% of OCR patients (4/18) had sufficient AB titers and significantly higher concentrations when compared to both T0 and T1 (Figure 4a).

Figure 4b summarizes the immunological effects of vaccinating mRNA and VVV non-responders with protein-based vaccines. Even after a single dose of NVX-CoV2373, 77% (14/18) of OCR and 33% (15/45) of S1PR, pwMS experienced sufficient cellular and/or humoral immunity (Figure 4b). After two doses, this increased to 88% (16/18) for OCR and 62% for S1PR pwMS. OCR pwMS were more likely to develop sufficient T cell responses after vaccination with NVX-CoV2373, whereas S1PR pwMS mostly developed humoral responses (Figure 4b).

## 4. Discussion

In our study, we aimed to elucidate the characteristics of SARS-CoV-2-specific T cells via the measurement of broader cytokine profiles and corresponding humoral response in pwMS on different DMTs after SARS-CoV-2 vaccination. In addition, we also assessed whether cytokines other than IFN-γ could characterize or even act as a marker of the T cell response, especially in patients who did not release IFN-γ. The evaluated cytokines can be classified into three broad groups: the proinflammatory Th-1 cytokines: IL-2, IFN-γ, and TNF-α; the counterbalancing Th-2 cytokines: IL-4, IL-5, IL-10, and IL-13; and the Th-17 cytokine IL-17A, which is actively involved in several autoimmune diseases including MS [21].

Our results show that the cytokine recall response in pwMS after SARS-CoV-2 vaccination is heavily Th-1-skewed, with a much smaller Th2 response, while IL-17A release remained minimal. IL-2, IFN-γ, and TNF-α were consistently released in much higher concentrations than the other cytokines. This corroborates previous reports which have shown that a Th-1 response to SARS-CoV-2 vaccines is elevated in pwMS and in healthy controls [12,14,22,23].

Our results also showed that in pwMS, IL-2 release had the strongest correlation among all other cytokines to IFN-γ release. In addition, IL-2 release was higher in patients who screened positive for IFN-γ release as opposed to those who did not, except in S1PR treated patients. IL-2 was even released in larger concentrations than IFN-γ upon stimulation with SARS-CoV-2 Ag. Studies in non-MS individuals also reported IL-2 release to be larger than IFN-γ release after vaccination, whereas during acute infection, a higher IFN-γ release is observed [24]. This is probably due to the different roles IFN-γ and IL-2 have in SARS-CoV-2 immunity, with IFN-γ being involved in viral clearance, and IL-2 playing a role in the longer-term memory cell recall responses [25,26]. It has even been recommended to carry out IL-2/IFN-γ dual measurement instead of IFN-γ alone to screen for T cell immunity after SARS-CoV-2 vaccination [24]. According to our results, dual measurement would also optimally screen for T cell immunity in pwMS not on S1PR therapy.

S1PR-treated patients had a significantly lower release of cytokines after two doses of SARS-CoV-2 mRNA/VVV. This cytokine release was not increased after receiving the booster vaccine, nor did receiving additional two doses of protein-based vaccines significantly bolster this response. S1PR modulators bind the S1PR in lymphocytes and prevent the egress of lymphocytes from the lymph nodes, thus reducing the amount of B and T cells available in the peripheral circulation and subsequently in the cerebrospinal fluid [11]. This mechanism of action is supposedly the reason why we observed reduced serum anti-RBD AB concentrations in S1PR patients after vaccination as compared to UT pwMS and GA patients, but higher concentrations as compared to OCR patients. It is also likely the cause of the nearly absent T cell cytokine response in S1PR patients.

In contrast, these cellular responses were also found to be significantly more robust for patients that are on aCD20 therapy. According to the literature, the humoral responses to SARS-Cov-2 mRNA vaccines on aCD20 therapy are directly correlated with the time since the last infusion; T cell responses, on the other hand, have been shown to remain robust or even increase after the infusion. In B-celldepleted pwMS, Apostolidis et.al showed a more robust antigen-specific CD8 T cell response induced after SARS-CoV-2 mRNA compared to healthy controls [14]. These findings could be corroborated by our group [13]. The reasons for such a robust CD8 response remain unclear. Previous reports showed increased gene expression in pro-inflammatory Th1 and myeloid cells 2 weeks and 6 months after aCD20 treatment [27]. Furthermore, it is hypothesized that with the depletion in B cells and the subsequent decrease in circulating immunoglobulins, more antigen is available for presentation to CD4+ and CD8+ T cells. This, in turn, could lead to larger concentrations of released cytokines [14]. Alternatively, the robust T cell response could be a compensatory mechanism due to the lack of circulating B cells [23]. We observed enhanced CD4 and CD8 T cellular responses in pwMS receiving vaccination at early time points after their last aCD20 cycle [13]. OCR patients consistently had the lowest percentage of seroconversion after primary immunization and booster, and the lowest concentration of anti-RBD ABs. This is likely due to B cell depletion since a reduced presence of detectable circulating B cells in the periphery has been shown to correlate with decreased seroconversion following SARS-CoV-2 vaccination in pwMS [28,29].

Interestingly, IL-5 release also correlated strongly with IFN-γ release and was significantly higher in patients who screened positive for IFN-γ release as compared to those who did not. However, its concentrations were much smaller than IFN-γ and IL-2. IL-5, which correlated more strongly to IFN-γ than IL-4, IL-10, and IL-13 characterize the small Th-2 response to SARS-CoV-2 vaccines in UT pwMS, as well as those under GA, and OCR treatment. For those under S1PR treatment, no Th-2 response was shown.

It has been reported in the literature that IL-13 plays a significant role in the recall response to SARS-CoV-2 vaccination in individuals without MS [30]. It has even been recommended in some studies to include IL-13 along with IFN-γ and IL-2 as a marker for T cell immunity against Sars-CoV-2. For instance, Kratzer et al. found IL-13, along with IFN-γ and IL-2, to be most discriminatory between vaccinated individuals and non-vaccinated healthy controls upon stimulation of whole blood with SARS-CoV-2 spike peptides [31]. This robust IL-13 response was not due to any pre-existing atopy. However, in our study involving pwMS, the IL-13 release was not so prominent. Its concentrations were similar to those of IL-5, another Th-2 cytokine, but the concentrations of IL-13 were much lower than those of IL-2 and IFN-γ. While IFN-γ, IL-2, and IL-5 all correlated strongly with IFN-γ release, IL-13 release only correlated moderately. In addition, mRNA and VVV non-responders who responded to two additional doses of protein-based vaccines showed statistically significant increases in concentrations of IFN-γ, IL-2, IL-5, and in some cases IL-4, but not of IL-13. This could be related to yet unknown immune mechanisms unique to MS or to technical differences in the culture and stimulation with SARS-CoV-2 antigens between our study and the other studies. More studies are required to elucidate the role of IL-13 in the recall response to SARS-CoV-2 vaccines in pwMS.

As mentioned previously, a relevant number of pwMS on S1PR modulators or OCR failed to develop a sufficient immune response after two or three doses of mRNA and/or VVV. These patients did, however, benefit from two additional doses of NVX-CoV2373 in our study. S1PR patients mostly presented an increased humoral response with no improvement in T cell and cytokine responses. Opposingly, OCR patients had a pronounced increase in IL-2 and IFN-γ and a smaller increase in IL-5 after both the first and second doses of the protein-based vaccine as compared to baseline. The concentrations of IL-10 and IL-13 did not increase significantly compared to baseline, despite their concentrations being similar to the levels observed in the fully vaccinated and booster cohorts. This cellular response consisting mainly of IL-2, IFN-γ, and, to a lesser extent, IL-5 is consistent with what we observed after primary immunization and after three doses of mRNA and VVV.

The reason why patients who did not respond to previous mRNA and VVV benefit from subsequent doses of protein-based vaccines remains unclear [32]. Since all vaccines in this study encode or contain the complete SARS-CoV-2 spike protein, the main difference between these different vaccine types is how the antigens are processed by the cells before presenting the antigen and triggering the immune responses. For BNT162b2, mRNA-1273, AZD1222, and Ad26.COV2.S, host cells must first transcribe and/or translate the genetic information for the spike protein before antigen presentation, whereas for NVX-CoV2373, the antigen must first be internalized, broken down, and then presented [33]. This may, in turn, lead to differences in the conformation of the spike protein between these vaccines and subsequently varying immunogenicity, which could cause the mRNA and VVV non-responders to develop immune responses to protein-based vaccines. Another hypothesis is that the root of this different immunogenicity lies in the non-antigen components of the different vaccines. While the other vaccines are unadjuvanted, NVX-CoV2373 contains Matrix M, an adjuvant containing saponin nanoparticles extracted from the tree Quillaja Saponaria [34]. This hypothesis is not very well supported by the existing literature, which does not conclusively show that adjuvanted vaccines against SARS-CoV-2 or other viruses result in better immune responses for pwMS treated with OCR or S1PR. In fact, one study showed decreased humoral responses for OCR patients in both adjuvanted and non-adjuvanted vaccines, while others showed either impaired or normal humoral responses for patients on S1PR therapy [33,35,36,37]. Studies specifically for Matrix M showed that this adjuvant increases the AB response as well as the Th-1 skewed T cell response [38,39]. This is consistent with the results of this study; however, further studies are needed to evaluate whether this immunogenicity is due to Matrix M or not. Another possibility is that the observed increase in immune responses upon protein-based vaccination is simply due to the repeated booster effect since we did not have a control group receiving fourth and fifth mRNA or VVV shots. Our results showed that pwMS on OCR and S1PR who did not respond to mRNA and VVV benefitted from additional doses of protein-based vaccines. We could not include healthy controls in this cohort due to the scarcity of healthy controls who are also non-responders to mRNA and VVV. We can assume, however, that the mechanism of this increased immunogenicity in non-responders to protein-based vaccines should be conserved in healthy controls. More studies on the matter are required to confirm/refute this assumption.

Our study bears some limitations. Of main concern is the lack of a healthy control group who would be vaccinated and then compared to the MS patients. This is why we carried out an extensive literature review on the immune responses of healthy controls to SARS-CoV-2 vaccinations. Furthermore, we do not have a comparator defined by unvaccinated MS patients. During the pandemic, based on ethical concerns, all pwMS were offered vaccination, and the study protocol defined a selection of already vaccinated patients per consecutive sampling. In our evaluations, we focused on the comparison of SARS-CoV2-specific T cellular cytokine profiles between different immunomodulatory treatment groups but were not able to find differences based on the unvaccinated antigen-specific cytokine profile of these patients. Another limitation of the study is that while assessing the difference in the immune response between the fully vaccinated and booster cohort, two different groups of pwMS were compared. A longitudinal cohort with the same patients similar to the one we used for the protein-based vaccine cohort would have been more suitable for the assessment of the changes in the immune response between the two groups. The main reason why this could not be achieved is that at the time of design of the study and during the early stages of its implementation (May 2021), booster vaccines were not yet authorized or recommended in Germany. Thus, this did not factor into the original study design. Beside vaccine-specific immunological characteristics between mRNA, viral vector and protein-based vaccines booster effects by additional vaccinations may also impact the degree of the SARS-CoV2-specific immune response. In Germany, a fourth or even more vaccinations have been recommended since 10/2022, when our studyhas already been completed. Further evaluations on additional booster vaccinations using mRNA versus protein-based vaccines could help to differentiate immunogenicity potential in different vaccines platforms.

## 5. Conclusions

In conclusion, this study comprehensively assesses cytokine profiles in pwMS subjected to four distinct therapeutic strategies in the context of mRNA, viral vector (VVV), and protein-based SARS-CoV-2 vaccination. Our findings indicate that pwMS exhibit a T-helper 1 (Th1)-biased cytokine response characterized by significant production of IFN-γ and IL-2, which parallels the profiles reported for healthy individuals following SARS-CoV-2 immunization. Additionally, a modest but detectable Th2 response was observed, predominantly marked by IL-5 production, with lesser contributions from IL-4, IL-10, and IL-13. For untreated pwMS and those receiving GA or OCR, supplementing conventional IFN-γ release assays with IL-2 and IL-5 measurements could be suggested to evaluate the cellular immune response to the various types of SARS-CoV-2 vaccines. Conversely, we were unable to identify a distinct cytokine signature in pwMS treated with S1PR modulators, as these patients consistently demonstrated an absence of cytokine release following antigenic stimulation by the SARS-CoV-2 antigen. Importantly, our study also revealed that pwMS non-responsive to two or three doses of mRNA or viral vector vaccines showed beneficial immune responses after receiving protein-based vaccines. This finding may carry significant implications for optimizing vaccine strategies in this patient population.

## Figures and Tables

**Figure 1 vaccines-12-00684-f001:**
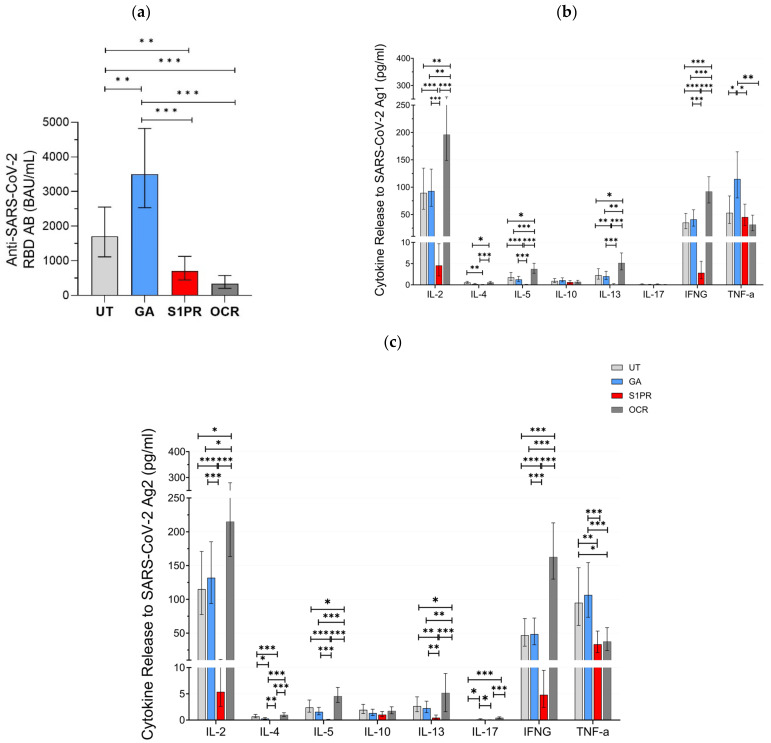
SARS-CoV-2-specific B and T cellular response in UT pwMS and pwMS treated with GA, S1PR, or OCR who received two doses of mRNA or VVV. (**a**) Anti-SARS-CoV-2 spike RBD AB titers are presented. The concentrations of different cytokines released after stimulation with SARS-CoV-2 Ag1 (**b**) and Ag2 (**c**) are demonstrated. Means with 95% confidence intervals are presented. Asterisks indicate level of statistical significance: * *p* < 0.05; ** *p* < 0.01; *** *p* < 0.001.

**Figure 2 vaccines-12-00684-f002:**
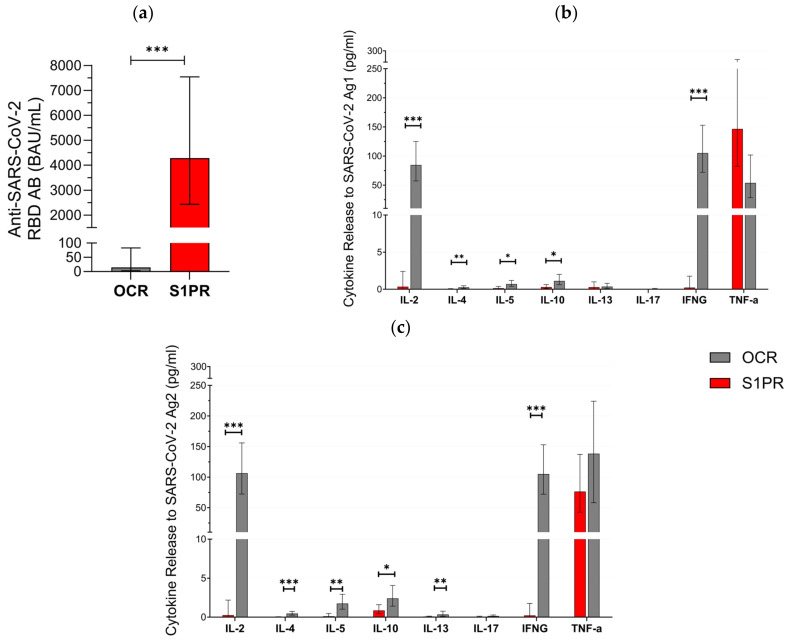
SARS-CoV-2-specific B and T cellular response in pwMS treated with S1PR or OCR who received booster vaccines doses of mRNA or VVV. (**a**) Anti-SARS-CoV-2 spike RBD AB titers are presented. The concentrations of different cytokines released after stimulation with SARS-CoV-2 Ag1 (**b**) and Ag2 (**c**) are demonstrated. Means with 95% confidence intervals are presented. Asterisks indicate level of statistical significance. * *p* < 0.05; ** *p* < 0.01; *** *p* < 0.001.

**Figure 3 vaccines-12-00684-f003:**
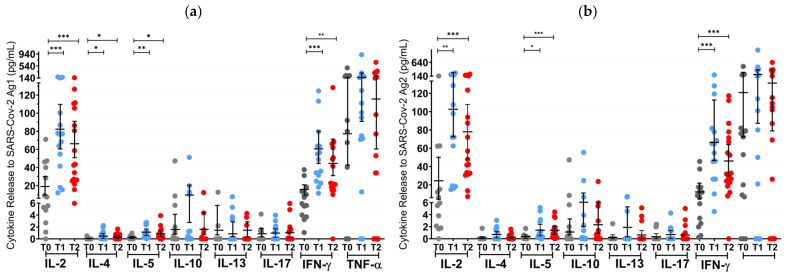
T cell cytokine profiles in mRNA and VVV non-responders after vaccination with NVX-CoV2373. Cytokine profile in OCR patients after stimulation with SARS-CoV-2 Ag1 (**a**) and Ag2 (**b**) as well as cytokine profile in S1PR patients after stimulation with SARS-CoV-2 Ag1 (**c**) and Ag2 (**d**) are presented. Scatter plots with means with 95% confidence intervals are presented for each time point per cytokine. T0, baseline measurement on the day of the first NVX-CoV2373 vaccination; T1, three weeks after first NVX-CoV2373 vaccination; T2, follow-up four to eight weeks after second vaccination with NVX-CoV2373. Asterisks indicate level of statistical significance: * *p* < 0.05; ** *p* < 0.01; *** *p* < 0.001.

**Figure 4 vaccines-12-00684-f004:**
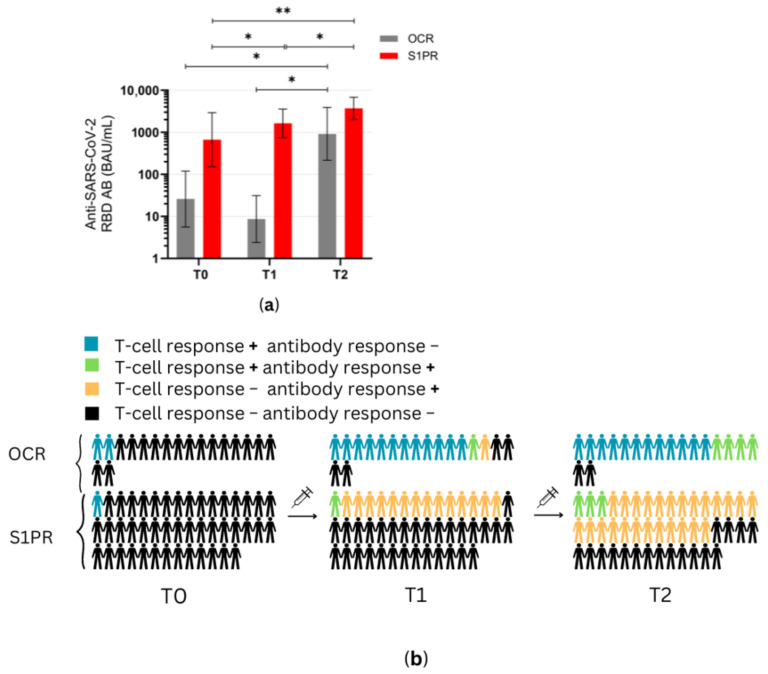
Antibody response profiles in mRNA and VVV non-responders after vaccination with NVX-CoV2373. (**a**) Anti-SARS-CoV-2 spike RBD AB titers of OCR and S1PR patients are presented with means with 95% confidence intervals in BAU/mL, log10. (**b**) Percentage of OCR versus S1PR patients that showed a positive (+) versus negative (−) T cell, B cell, or T and B cell response after NVX-CoV2373. T0, baseline measurement on the day of the first NVX-CoV2373 vaccination; T1, three weeks after first NVX-CoV2373 vaccination; T2, follow-up four to eight weeks after second vaccination with NVX-CoV2373. Asterisks indicate level of statistical significance: * *p* < 0.05; ** *p* < 0.01.

**Table 1 vaccines-12-00684-t001:** Patient characteristics of the fully vaccinated and booster cohorts.

Characteristic	Fully Vaccinated Cohort (*n* = 126)	Booster Cohort (*n* = 28)
Sex, n (%)		
Female	93 (73.8)	18 (64.3)
Age, years		
Mean ± sd	48.37 ± 12.32	50.68 ± 9.9
Median	48	49
Range	21-77	32-68
MS subtype, n (%)		
CIS	2 (1.6)	0 (0.0)
RRMS	97 (75.8)	17 (60.7)
PPMS	13 (10.2)	3 (10.7)
SPMS	16 (12.5)	8 (28.6)
DMT modality		
UT	28 (22.2)	-
GA	27 (21.1)	-
OCR	39 (30.5)	16 (57.1)
S1PR	32 (25.4)	12 (42.9)
Treatment duration, days: mean ± SD		
GA		
OCR	3440 (2510)	
Time since last infusion	912 (616)	1295 (762)
S1PR	157 (124)	133 (130)
	1440 (1321)	1180 (1080)
Time vaccination–sampling		
days, mean ± sd	63.75 ± 39.1	74.96 ± 44.1
days, range	6-203	12-147
Previous COVID-19 Infections, n (%)		
Yes	10 (7.9)	8 (28.6)
No	116 (92.1)	20 (71.4)
Vaccines, n (%)	2x mRNA, 114 (90.4)	3x mRNA, 22 (78.6)
	2x VVV, 6 (4.8)	2 mRNA + 1 VVV, 5 (17.9)
	1x mRNA + 1 VVV, 6 (4.8)	1x mRNA + 2 VVV, 1 (3.6)

CIS = clinically isolated syndrome; GA = glatiramer acetate; mRNA = messenger ribonucleic acid; OCR = Ocrelizumab; PPMS = primary progressive multiple sclerosis; RRMS = relapsing remitting multiple sclerosis; S1PR = sphingosine-1 phosphate receptor modulators; SPMS = secondary progressive multiple sclerosis; UT = untreated; VVV = viral vector vaccine.

**Table 2 vaccines-12-00684-t002:** Patient characteristics the protein-based vaccination subgroup.

Characteristic	Protein-Based Vaccine Cohort (*n* = 63)
Sex, n (%)	
Female	33 (52.38)
Age, years	
Mean ± sd	49.74 ± 11.22
Median	51
Range	24–72
MS subtype, n (%)	
RRMS	53 (84.1)
PPMS	2 (3.2)
SPMS	8 (12.7)
DMT modality	
OCR	18 (28.57)
S1PR	45 (71.43)
Treatment duration, days: mean ± SD	
OCR	901 (462)
Time since last infusion	96.61 (69.29)
S1PR	2201 (1096)
Previous vaccines, n (%)	
2x mRNA/VVV	2 (3.2)
3x mRNA/VVV	59 (93.6)
COVID-19 + 2x mRNA/VVV	2 (3.2)

mRNA = messenger ribonucleic acid; OCR = Ocrelizumab; PPMS = primary progressive multiple sclerosis; RRMS = relapsing remitting multiple sclerosis; S1PR = sphingosine-1 phosphate receptor modulators; SPMS = secondary progressive multiple sclerosis; VVV = viral vector vaccine.

## Data Availability

The dataset will be made available by the authors upon request.

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
