# Peer review of "SARS-CoV-2-Specific Immune Cytokine Profiles to mRNA, Viral Vector and Protein-Based Vaccines in Patients with Multiple Sclerosis: Beyond Interferon Gamma"

_vaccines, 2024, doi:10.3390/vaccines12060684_

Round 1
Reviewer 1 Report
Comments and Suggestions for Authors
Shows that in response to COVID vaccinations, glatiramer decreases T > B responses; ocrelizumab decreases B > T. Clearly written but could be condensed. Minor changes suggested.
Abstract:
13 T cells no hyphen
14 “release of” – is this in serum or culture?
19 “predominant” is vague
Introduction:
79 define what LIASON does
Results:
190 Clarify when tested with Ocre vs. last shot. The temporal gap is unclear.
215 What is the range for the non-response interval?
236 Add times for T1 and T 2 here also
252 at T2, 4-8 weeks, no…
Discussion
283 cytokines, IL-2
285, 315 never use “known to be “ use “which is actively…”
322 increased responses soon after infusion is puzzling and does not make biological sense. For instance, C Fong, X Feng et al., J Neuroimmunology 2023 or 2024 found Ig mRNA and T cell and monocyte activation rose most near the pre-infusion point. Nonetheless. C Woopen NNINI 2024 replicated the early T cell response that was found in the current paper.
332 “except for those… is unclear.
339 was not due to
This paragraph is wordy
383 What is Matrix M?
389 onward clarify cytokine levels vs. time after infusion
Figures:
2, 4. use same colors and sequence as in Fig 1, and also within Fig 2.
4 b increase font; bigger or bolder
S1 better to cluster Th1, then Th2 together to illustrate inflammatory/regulatory profiles.
Define S1, S2
Also R upper quadrant could mirror LLQ with p-values in RUQ.
Author Response
Dear Reviewer 1, thank you very much for your critical review of the manuscript. Please find attached our response, to our comments:
Comments: Abstract:
13 T cells no hyphen
14 “release of” – is this in serum or culture?
19 “predominant” is vague
Response: We corrected the suggested parts.
Comments: Introduction:
79 define what LIASON does
Response: We defined LIASON in the manuscript.
Results:
Comment: 190 Clarify when tested with Ocre vs. last shot. The temporal gap is unclear.
Response: Now included in Table 1.
Comments:
215 What is the range for the non-response interval?
236 Add times for T1 and T 2 here also
252 at T2, 4-8 weeks, no…
Response: We added and corrected the suggested parts.
Discussion
Comments:
283 cytokines, IL-2
285, 315 never use “known to be “ use “which is actively…”
Response: We added and corrected the suggested parts.
Comment: 322 increased responses soon after infusion is puzzling and does not make biological sense. For instance, C Fong, X Feng et al., J Neuroimmunology 2023 or 2024 found Ig mRNA and T cell and monocyte activation rose most near the pre-infusion point. Nonetheless. C Woopen NNINI 2024 replicated the early T cell response that was found in the current paper.
Response: We added a further discussion on that in the manuscript.
Comments:
332 “except for those… is unclear.
339 was not due to
This paragraph is wordy
383 What is Matrix M?
Response: We added and corrected the suggested parts.
Comment: 389 onward clarify cytokine levels vs. time after infusion
Response: Time between last infusion and vaccination was added in the manuscript.
Comments: Figures:
2, 4. use same colors and sequence as in Fig 1, and also within Fig 2.
4 b increase font; bigger or bolder
S1 better to cluster Th1, then Th2 together to illustrate inflammatory/regulatory profiles.
Define S1, S2
Also R upper quadrant could mirror LLQ with p-values in RUQ.
Response: We added and corrected the suggested parts.
Reviewer 2 Report
Comments and Suggestions for Authors
I thank the authors for their work. The study of the immune response in patients with MS is of significant scientific interest and may have implications for therapy. Unfortunately, the authors do not pay enough attention to the control group. Please report data for unvaccinated MS patients in all experiments.
Lines 60-85. The distribution of patients into groups is not described clearly enough. first, second and third cohorts combine patients from different groups. This is not true. The control group should be allocated separately. Patients receiving different treatments should also be considered separately.
Line 95. “For patients in the fully vaccinated cohort.” Why wasn't the study done on a control group of unvaccinated patients?
Line 145-146 “our participants were previously infected with SARS-CoV-2 prior to sampling.” By what method was previous infection with SARS-CoV-2 determined? The infection may be asymptomatic and patients should therefore be tested for antibodies to the N protein. It is known that vaccinated patients have antibodies only to the S protein, but not to other proteins of this coronavirus.
Author Response
Dear Reviewer 2, thank you very much for your critical review of the manuscript. Please find attached our response, to our comments:
Comment: I thank the authors for their work. The study of the immune response in patients with MS is of significant scientific interest and may have implications for therapy. Unfortunately, the authors do not pay enough attention to the control group. Please report data for unvaccinated MS patients in all experiments.
Response: Unfortunately, we do not have a comparator defined by unvaccinated MS patients. During the pandemic only selected patients refused SARS-CoV2 vaccination at our center and these patients were mostly not willing to attend in the study. Furthermore it was not ethically acceptable patients not to recommend the vaccination. Based on the study protocol patients were asked to be included in the study after they were already vaccinated. There was a consecutive sampling. The study was designed to evaluate difference in cytokine profile after SARS-CoV2 vaccination between the different available treatment options and their mechanism of action, as these aspect seems to be the most relevant aspect and difference regard SARS-CoV2 specific immune responses after vaccination in MS patients. We included a limitation part on that in the discussion.
Comment: Lines 60-85. The distribution of patients into groups is not described clearly enough. first, second and third cohorts combine patients from different groups. This is not true. The control group should be allocated separately. Patients receiving different treatments should also be considered separately.
Response: During the study a consecutive sampling in a real world setting was performed. Three different cohorts/approaches were defined including (1) fully vaccinated group, (2) booster group and (3) protein based vaccinated group. In these recruited cohorts different treatment scenarios were separated. Based on the ongoing changing recommendations during the pandemic, patients with no or low immune response to the SARS-CoV2 vaccination were boostered and received more vaccinations than other treatment groups. Due to ethical issues patients were free to receive further booster vaccinations as recommended. As the untreated MS patient group is part of the fully vaccinated cohort we would not recommend to split that. The aim of the study was to demonstrate differences regard SARS-CoV2 specific immune responses after vaccination in different treatment scenarios but not differences in the clinical efficacy between the treatment groups.
Comment: Line 95. “For patients in the fully vaccinated cohort.” Why wasn't the study done on a control group of unvaccinated patients?
Response: as already mention above: Unfortunately, we do not have a comparator defined by unvaccinated MS patients. During the pandemic only selected patients refused SARS-CoV2 vaccination at our center and these patients were mostly not willing to attend in the study. Furthermore it was not ethically acceptable patients not to recommend the vaccination. Based on the study protocol patients were asked to be included in the study after they were already vaccinated. There was a consecutive sampling. The study was designed to evaluate difference in cytokine profile after SARS-CoV2 vaccination between the different available treatment options and their mechanism of action, as these aspect seems to be the most relevant aspect and difference regard SARS-CoV2 specific immune responses after vaccination in MS patients. We included a limitation part on that in the discussion.
Comment: Line 145-146 “our participants were previously infected with SARS-CoV-2 prior to sampling.” By what method was previous infection with SARS-CoV-2 determined? The infection may be asymptomatic and patients should therefore be tested for antibodies to the N protein. It is known that vaccinated patients have antibodies only to the S protein, but not to other proteins of this coronavirus.
Response: SARS-CoV2 infections confirmed by PCR or COVID-19 Antigen Rapid Test Kit were included in our evaluation (details on that now included in the manuscript). However, we cannot exclude asymptomatic infections in selected patients. At time point when study was performed as well as today it is not concluding clear if titer of N protein are influenced by selected immunomodulatory therapies in MS patients and so robust enough to define asymptomatic infections in all patients. Association of SARS-CoV2 infections and T as well as B cellular response in vaccinated patients was only a secondary endpoint and not the main focus of the study.
Reviewer 3 Report
Comments and Suggestions for Authors
SARS-CoV-2-specific Immune Cytokine Profiles to mRNA, Viral Vector and Protein-based Vaccines in Patients with Multiple Sclerosis: Beyond Interferon Gamma
This article characterizes the cellular immune response to SARS-CoV-2 vaccines in pwMS. The study findings indicate that immune responses to SARS-CoV-2 vaccines in pwMS are predominantly skewed towards a Th1 phenotype, characterized by IL-2 and IFN-γ. Additionally, a Th2 response characterized by IL-5 and, to a lesser extent, IL-4, IL-10, and IL-13 are observed. Authors suggested that measuring IL-2 and IL-5 levels could complement traditional IFN-γ assays to characterize the cellular responses to SARS-CoV-2 vaccines. The study provides a comprehensive cytokine profile for pwMS receiving different DMTs and offers insights for designing vaccination strategies. The study was meticulously designed, and the experiments were performed well. The manuscript is well written, the following are the specific comments to strengthen the manuscript further,
1. In Figure 1, consider enhancing the resolution of graphs, specifically the statistics portion.
2. Also, consider using clear, visible asterisks in Figure 2. Also, all other figures should be regarded for better visualization.
3. Thus, the authors suggested that protein-based SARS-CoV-2 is better than mRNA vaccines. Is this specifically for pwMS, or can it be extrapolated for the general population?
4. As booster cohorts are now available, what could be the results if these booster cohorts were included in the study?
Author Response
Dear Reviewer, thank you very much for your critical review of the manuscript. Please find attached our response, to our comments:
Comment1+2. In Figure 1, consider enhancing the resolution of graphs, specifically the statistics portion. 2. Also, consider using clear, visible asterisks in Figure 2. Also, all other figures should be regarded for better visualization.
Response: Thank you for our comment. We reworked the graphs to improve presentation.
Comment 3.: Thus, the authors suggested that protein-based SARS-CoV-2 is better than mRNA vaccines. Is this specifically for pwMS, or can it be extrapolated for the general population?
Response: Our results showed that pwMS on OCR and S1PR who did not respond to mRNA and VVV benefitted from additional doses of protein-based vacccines. We could not include healthy controls in this cohort due to the scarcity of healthy controls who are also non-responders to mRNA and VVV. We can assume however, that the mechanism of this increased immunogenicity in non-responders to protein-based vaccines should be conserved in healthy controls. More studies on the matter are required to confirm/refute this assumption. We added a discussion on that in the manuscript.
Comment 4.: As booster cohorts are now available, what could be the results if these booster cohorts were included in the study?
Response: In our study we included evaluations on first booster vaccination with mRNA or protein-based vaccines as well as second booster vaccination with protein based vaccines. However we have no data on further booster vaccinations as these were not available at time point when study was performed. We added a discussion on that in the manuscript.
Reviewer 4 Report
Comments and Suggestions for Authors
Disease-modifying therapies (DMTs) affect the immune response to SARS-CoV-2 vaccines in patients with multiple sclerosis (pwMS). This study aimed to detail the characteristics of antigen-specific T-cells by measuring various cytokines in pwMS on different DMTs. Authors assessed the SARS-CoV-2-specific T-cell responses through the release of cytokines IL-2, IL-4, IL-5, IL-10, IL-13, IL-17A, IFN-γ, and TNF-α, as well as antibodies against the SARS-CoV-2 spike protein, following two or three doses of mRNA or viral vector vaccines (VVV). Non-responders to mRNA vaccines received the NVX-CoV2373 protein-based vaccine, and their immune responses were evaluated. The study found that pwMS predominantly exhibited a Th1 immune response, characterized by IL-2 and IFN-γ, and also a Th2 response, marked by IL-5 and, to a lesser extent, IL-4, IL-10, and IL-13.
The paper is well-designed and executed. The statistical analysis is appropriate. The presentation of the results is clear and concise, with figures of high quality that effectively illustrate the findings. The authors provide a thorough discussion of the background, contextualizing their study within the existing literature, and offer insightful comments on the results. Overall, the paper is of good quality and merits publication in its current form.
Author Response
Dear Reviewer, thank you very much for the critical review of our manuscript and the positive comments.
Round 2
Reviewer 2 Report
Comments and Suggestions for Authors
The new additions to the manuscript made a big difference. The quality of the paper had improved, and all my questions were addressed. No more comments.